# MMP-12 Inhibitors Inverse Eosinophilic Inflammation-Mediated Bronchial Fibrosis in Murine Models of Pulmonary Airway Obstruction

**DOI:** 10.3390/cells14171307

**Published:** 2025-08-23

**Authors:** Chandra Sekhar Kathera, Chandra Sekhar Yadavalli, Anil Mishra

**Affiliations:** John W. Deming Department of Medicine, Tulane Eosinophilic Disorders Center (TEDC), Section of Pulmonary Diseases, Tulane University School of Medicine, New Orleans, LA 70112, USA; ckathera@tulane.edu (C.S.K.); cyadavalli@tulane.edu (C.S.Y.)

**Keywords:** *Aspergillus fumigatus*, IL-13, bronchial fibrosis, EMT, MMP-12

## Abstract

Matrix metalloproteinases (MMPs) are a major group of proteases known to regulate the turnover of the extracellular matrix (ECM). We observed that induced MMP-12 promotes eosinophilic inflammation-related epithelial cell mesenchymal transition (EMT), bronchial fibrosis, and airway obstruction in an allergen-exposed mouse model of chronic airway diseases in allergen-exposed mice and in airway-specific CC10-IL-13-overexpressed mice. Our histological analysis showed that the parabronchial and perivascular accumulation of eosinophils, fibroblasts, and collagen is significantly decreased in MMP-12^−/−^ allergen-exposed mice and airway-specific rtTA-MMP-12^−/−^CC-10-IL-13-overexpressed mice compared to allergen-exposed wild-type mice and rtTA-CC10-IL-13-overexpressed mice. ELISA and Western blot analyses validated these histological findings, demonstrating that EMT and profibrotic protein levels were significantly decreased in allergen-challenged MMP-12^−/−^ mice and rtTA-MMP-12^−/−^CC10-IL-13-overexpressed mice in comparison to the allergen-exposed wild-type mice and rtTA-CC10-IL-13-overexpressed mice. In addition, we also observed that allergen-challenged MMP-12^−/−^ mice have improved resistance and compliance compared to allergen-challenged wild-type mice. Most importantly, we show that treatment with MMP-12 inhibitors (PF-00356231 and MMP408) restricts the induction and progression of bronchial fibrosis and airway restrictions in allergen-exposed mice and airway-specific rtTA-CC10-IL-13 mice compared to the respective control mice. Taken together, the novelty of these findings lie in the fact that induced MMP-12 regulates eosinophilic inflammation-induced bronchial fibrosis and associated airway restriction, which may be reduced by treatment with MMP-12 inhibitors.

## 1. Introduction

Airway inflammation, a hallmark of respiratory diseases like asthma and chronic obstructive pulmonary disorder (COPD), is characterized by the presence of inflammatory cells that lead to recurring symptoms and reversible airflow obstruction. These conditions are on the rise worldwide, including in the USA, affecting millions and causing thousands of deaths annually (http://www.aafa.org). Eosinophilic bronchial pathogenesis, including fibrosis, is highly heterogeneous and caused by diverse factors, such as infections, genetics, and inflammation, and often overlaps with more common disorders triggered by environmental pollution, like cigarette smoke, or drug toxicity. This complexity often leads to a costly and lengthy diagnostic period. 

A key factor in the progression of chronic airway diseases is pulmonary airway remodeling, a process that involves the tightening of muscles around the airways, making it harder to breathe. This remodeling is driven by the matrix metalloproteinase (MMP) family of enzymes, which are crucial to tissue remodeling and repair [1,2] and play significant roles in COPD and asthma pathogenesis [1,2,3,4,5,6,7]. The precise functions of MMP-2, MMP-9, and MMP-12 have been elegantly demonstrated in a model of allergic airway inflammation in mice [8]. MMP-12, known as a macrophage metalloelastase, is particularly notable due to its association with tissue destruction and aberrant repair processes [9]. It is also found at high levels in patients with inflammatory diseases, including pulmonary pathogenesis [10,11]. Mice deficient in MMP-12 exhibit a marked reduction in airway eosinophilic inflammation after an allergic injury [11]. Additionally, some evidence suggests that inflammation triggered by the cytokine IL-13 may lead to MMP-mediated emphysema [12] and promote tissue remodeling and fibrosis [10,11]. 

Despite significant advancements in our understanding of how eosinophil-related bronchial inflammation leads to fibrosis and functional abnormalities, effective therapeutic strategies are limited. This highlights the need to further explore the role of MMP-12 in allergen-induced bronchial fibrosis and airway obstruction. 

To address this gap, we investigated the mechanisms of MMP-12 in eosinophilic inflammation-induced fibrosis and associated functional abnormalities using allergen- and IL-13-induced mouse models of asthma. The current study shows that mice challenged with *Aspergillus* were protected from inflammation-induced changes, including increased levels of proteins associated with fibrosis and airway resistance, when treated with MMP-12 inhibitors. This suggests that targeting MMP-12 could be a novel strategy for controlling allergen-induced eosinophilic inflammation and its associated airway diseases. These findings have the potential to revolutionize our understanding of how MMP-12 contributes to bronchial fibrosis. They also provide a method for developing single, targeted therapeutic interventions to reverse the progression of established fibrosis and airway obstruction.

## 2. Materials and Methods

### 2.1. Animals

IL-13 was conditionally expressed in the lung airways of transgenic mice using the DOX-inducible Clara cell 10 kDa (CC10) promoter. MMP-12 gene-deficient (MMP-12^−/−^) mice and their wild-type (WT) counterparts were obtained from Jackson Laboratories (Bar Harbor, ME, USA). To generate MMP-12-deficient IL-13-overexpressing mice, MMP-12^−/−^ background mice were crossed with lung (airway)-specific CC10–rtTA-IL-13 mice. The CC10–rtTA-IL-13 mice used were obtained from the laboratory of Dr. Marc Rotheberg, Cincinnati Children’s Hospital, Cincinnati, OH, and originally generated by Dr. Elias of Yale University [12]. The desired MMP-12^−/−^CC10–rtTA-IL-13 progeny were identified in the F2 generation. Genotypes of all progenies were confirmed by polymerase chain reaction (PCR). All animal procedures were approved by the Institutional Animal Care and Use Committee and conducted in accordance with the National Institutes of Health guidelines for the care and use of laboratory animals.

### 2.2. Mouse Treatments

Eight-week-old WT, MMP-12^−/−^, CC10–rtTA-IL-13, and MMP-12^−/−^CC10–rtTA-IL-13 bi-transgenic mice were used for all experiments. Mice were anesthetized with isoflurane and treated intranasally with either phosphate-buffered saline (PBS) or *Aspergillus fumigatus* (*Af*) extract (100 μg) three times per week for 3 weeks. To induce IL-13 expression, bi-transgenic mice were fed either doxycycline (DOX)-supplemented chow or regular chow for the same period. In select groups, mice received intraperitoneal injections of MMP-12 inhibitors (PF-00356231, Cat.no.: HY-114091, MedChem Express, Monmouth Junction, NJ, USA; MMP408, Cat. No.: 444291; MilliporeSigma, Burlington, MA, USA) at a dose of 3.5 mg/kg/week [13]. Mice were euthanized and analyzed 20–24 h after the final allergen challenge.

### 2.3. Histopathology and Immunohistochemistry

Lung tissues were fixed, paraffin-embedded, and sectioned at 5 µm. Sections were deparaffinized with xylene, rehydrated through graded ethanol, and washed in PBS. Endogenous peroxidase activity was quenched using 3% hydrogen peroxide in methanol for 20 min. Non-specific binding was blocked with 3% goat serum for 1 h at room temperature. Sections were incubated overnight at 4 °C with primary antibodies targeting MMP-12 (Proteintech Group, Rosemont, IL, USA), TGF-β, vimentin, E-cadherin, N-cadherin, MUCIN, MUC-5AC, EPX, and FSP-1 (Santa Cruz Biotechnology, Dallas, TX, USA), as well as SMAD-4 and α-SMA (Cell Signaling Technology, Danvers, MA, USA). The next day, sections were incubated with biotinylated secondary antibodies for 1 h, developed using DAB substrate (Vector Laboratories, Newark, CA, USA), and counterstained with nuclear fast red. Slides were mounted and visualized under an Olympus BX43 optical microscope (Tokyo, Japan). Two investigators, blinded to the treatment groups, captured and quantified the images using Olympus CellSens Dimension software 4.2 version [14].

### 2.4. Collagen Staining and Quantification

Collagen deposition was assessed using Masson’s trichrome staining (Poly Scientific, Dalllas, TX, USA) following the manufacturer’s instructions. Stained sections were visualized with an Olympus BX43 microscope, and the collagen-positive area (expressed as square microns) was quantified using CellSens Dimension software 4.2 version [14].

### 2.5. Western Blot Analysis

Lung tissues were homogenized in protein extraction reagent (Thermo Scientific, San Diego, CA, USA) containing protease and phosphatase inhibitor cocktails (Sigma-Aldrich, Burlington, MA, USA). Protein concentrations were determined, and equal amounts (20 µg) were resolved by SDS-PAGE using 4–15% TGX gels (Bio-Rad, Hercules, CA, USA) and transferred onto PVDF membranes (MilliporeSigma, Burlington, MA, USA). Membranes were probed with primary antibodies against MMP-12; Collagen I, III, and IV (Col I, Col III, Col IV); fibroblast-associated growth factor and protein (FGF and FSP-1, respectively); profibrotic cytokines and signaling molecules (TGF-β, α-SMA, SMAD3, SMAD4, fibronectin); and epithelial cell mesenchymal transition (EMT)-associated proteins (E-cadherin, N-cadherin, vimentin, snail). GAPDH was used as a loading control. Antibodies were detected using standard enhanced chemiluminescence. The specific antibodies used, their dilutions for both Western blot and immunohistochemistry, and their manufacturers are provided in Appendix A.

### 2.6. ELISA Analysis

Cytokine and chemokine levels (IL-4, IL-5, IL-13, IL-18, VIP, eotaxin-I, eotaxin-II, and TGF-β1) were measured in serum and lung homogenates using DuoSet ELISA kits (Thermo Fisher Scientific, San Diego, CA, USA) according to the manufacturer’s protocols. 

### 2.7. Airway Resistance Measurement

Following the surgical exposure of the trachea in anesthetized mice, a cannula was meticulously inserted and secured with a ligature to prevent air leakage. Subsequently, mice were positioned within a Buxco Finepointe RC system chamber (Data Sciences International [DSI], St. Paul, MN, USA) and connected to a mechanical ventilator. Mechanical ventilation was then initiated at a predetermined respiratory rate and tidal volume. After acquiring two 3 min baseline measurements, mice were exposed to either PBS or methacholine (Cat.No.: A2251, Sigma-Aldrich, Burlington, MA, USA) at concentrations ranging from 3.125 to 50 mg/mL [15]. Airway resistance (RI) was then quantified according to the manufacturer’s specified protocol.

### 2.8. MMP-12 Inhibitors

MMP408 (Cat. No.: 444291; MilliporeSigma, Burlington, MA, USA) is a specific inhibitor for MMP-12 [16]. PF-00356231-hydrochloride inhibits not only MMP-12 but also MMP-2 and MMP-9. Both inhibitors are absorbed into the bloodstream after administration and are primarily excreted, along with their metabolites, through the kidneys. The treatment dose was selected based on a previously reported study [17].

### 2.9. Statistical Analysis

All data are presented as mean ± SEM. Statistical analysis was performed using GraphPad Prism 10.5 (GraphPad Software, San Diego, CA, USA). For comparisons between two groups, an unpaired two-tailed Student’s *t*-test was used. For comparisons involving more than two groups, a one-way analysis of variance (ANOVA) was performed, followed by Dunn’s or Tukey’s multi-group comparisons test. A *p*-value of < 0.05 was considered statistically significant. All data were confirmed to follow a normal distribution (*p* > 0.05). The detailed results for each figure can be found in the Appendix A.

## 3. Results

### 3.1. MMP-12 Regulates Eosinophil-Responsive Proinflammatory Cytokines and Promotes Epithelial–Mesenchymal Transition in Aspergillus fumigatus Extract-Challenged Mice

We investigated whether MMP-12 contributes to eosinophilic inflammation-induced EMT, a process that leads to bronchial fibrosis, in a mouse model of experimental asthma. Following an *Af* aeroallergen challenge protocol (Figure 1A), immunohistochemical analyses of lung sections revealed significantly reduced eosinophilic infiltration (anti-EPX^+^) in MMP-12^−/−^ mice compared to WT mice (Figure 1B). Further analysis revealed that the expression of EMT markers—vimentin, N-cadherin, and snail—was markedly decreased in *Af-*challenged MMP-12^−/−^ mice relative to *Af-*challenged WT mice (Figure 1C–E). Western blotting validated these findings and offered additional insight, showing reduced total proteins levels of the EMT markers in *Af-*challenged MMP-12^−/−^ mice. This was accompanied by elevated E-cadherin levels (Figure 1F), including MMP-12, indicating a reversal of the EMT process. Additionally, ELISA revealed lower levels of eosinophil-responsive chemokines (eotaxin-1, eotaxin-2, VIP) and proinflammatory cytokines (IL-4, IL-5, IL-18) in *Af-*challenged MMP-12^−/−^ mice compared to *Af-*challenged WT mice (Figure 1G,H). Both MMP-12^−/−^ and WT mice challenged with saline showed comparable baseline levels of these markers. These data suggest that MMP-12 plays a central role in regulating eosinophilic inflammation and associated EMT proteins during allergic airway inflammation. Detailed histological and morphometric analyses can be found in Appendix A. Photomicrographs are presented at 400× original magnification. All data are shown as mean ± SEM, with *n* = 3 experiments and four mice per group per experiment. 

### 3.2. MMP-12 Drives Fibroblast Accumulation and Fibrosis in Aspergillus fumigatus Extract Allergic Airway Inflammation

Given the reduced EMT- and fibrosis-associated proteins observed in *Af-*challenged MMP-12^−/−^ mice, we next examined whether MMP-12 influences fibroblast recruitment and profibrotic cytokines. Immunohistochemistry revealed a marked accumulation of MMP-12-positive (MMP-12^+^) cells in the lung airways of *Af-*challenged WT mice, which was absent in *Af-*challenged MMP-12^−/−^ mice (Figure 2A). Several key profibrotic markers were also significantly reduced in the lungs of *Af-*challenged MMP-12^−/−^ mice compared to *Af-*challenged WT mice. These include FSP-1, TGF-β, α-SMA, SMAD4, and peri-bronchial and perivascular collagen (Figure 2B–F). Western blotting confirmed these findings and offered the additional insight of a decrease in total protein expression for fibroblast and fibrosis markers (FSP-1, Col I/IV, TGF-β, α-SMA, SMAD4, MMP-12, and FGF) in *Af-*challenged MMP-12^−/−^ mice (Figure 2G). An ELISA for TGF-β also showed a significant reduction in *Af-*challenged MMP-12^−/−^ mice compared to *Af-*challenged WT mice (Figure 2H). These findings confirm that MMP-12 is a critical regulator of allergen-induced fibroblast accumulation, which promotes airway fibrosis. Detailed histological and morphometric analyses are presented in Appendix A. All photomicrographs are presented at 400× original magnification. Data are expressed as mean ± SEM, with *n* = 3 experiments and four mice per group per experiment.

### 3.3. MMP-12 Deficiency Improves Airway Functional Abnormalities in Aspergillus fumigatus-Challenged Mice

We next assessed how MMP-12 impacts airway pathology following an *Af* challenge by examining mucus-producing cells and lung function in MMP-12^−/−^ and WT mice. PAS staining revealed significantly fewer mucus-producing goblet cells in *Af-*challenged MMP-12^−/−^ mice than in the *Af-*challenged WT mice (Figure 3A, i–iii). Immunostaining and morphometric analysis confirmed this reduction, showing lower levels of MUCIN-1 and MUC-5AC (Figure 3B,C). Pulmonary function testing using the Buxco Finepointe system showed lower airway resistance (Figure 3D) and increased compliance (Figure 3E) in *Af-*challenged MMP-12^−/−^ mice compared to *Af-*challenged WT mice, suggesting that the absence of MMP-12 helps preserve lung function. Saline-challenged mice in both groups showed comparable baseline airway resistance and compliance, indicating that MMP-12 deficiency alone does not affect lung function (Figure 3D,E). We also found that IL-13, a key mediator of airway goblet cell hyperplasia and remodeling [18], was reduced in *Af-*challenged MMP-12^−/−^ mice compared to *Af-*challenged WT mice. Interestingly, even saline-treated MMP-12^−/−^ mice showed lower IL-13 levels than their WT counterparts, suggesting that MMP-12 deficiency may affect T cell function (Figure 3F). These findings indicate that MMP-12 regulates the IL-13-dependent impairment of lung function in eosinophilic mice. Detailed histological and morphometric analyses are provided in Appendix A. All photomicrographs are presented at 400× original magnification. Data are expressed as mean ± SEM, with *n* = 3 experiments and four mice per group per experiment.

### 3.4. MMP-12 Regulates IL-13 Overexpression-Induced Bronchial Fibrosis

To determine whether IL-13-induced fibrosis is dependent on MMP-12, we generated MMP-12^−/−^CC10–rtTA-IL-13 mice by cross-breeding MMP12-deficient mice (MMP-12^−/−^) with lung (airway)-specific IL-13 overexpressing mice (CC10–rtTA-IL-13). We exposed both the generated line (MMP-12^−/−^CC10–rtTA-IL-13) and a control group (MMP^+/+^CC10–rtTA-IL-13) to DOX food or no-DOX food for 3 weeks to induce IL-13 expression in the lungs, as shown in our protocol (Figure 4A). We then analyzed lung tissues for markers of fibrosis (collagen deposition using Masson’s trichrome staining; TGF-β and α-SMA immunostaining) and found that, compared to DOX-fed MMP^+/+^CC10–rtTA-IL-13 mice, MMP-12^−/−^CC10–rtTA-IL-13 mice exhibited significantly reduced peribranchial and perivascular collagen deposition (Figure 4B), reduced α-SMA expression (Figure 4C), and fewer TGF-β-positive cells (Figure 4D). Morphometric analysis confirmed significantly reduced fibrosis and profibrotic cytokines in DOX-fed MMP-12^−/−^CC10–rtTA-IL-13 mice (Figure 4B–D, iii). Western blot analysis validated these findings, showing a significant decrease in the expression of profibrotic proteins, including Col I/III, TGF-β, SMAD3/4, α-SMA, and fibronectin, in DOX-exposed MMP-12^−/−^CC10–rtTA-L-13 mice compared to MMP^+/+^CC10–rtTA-IL-13 control mice (Figure 4E). Densitometry showed no change in SMAD3, but a significant change in SMAD4. ELISA analysis showed significantly reduced IL-13 levels in DOX-exposed MMP-12^−/−^CC10–rtTA-IL-13 mice compared to MMP^+/+^CC10– rtTA-IL-13 mice (Figure 4F), whereas comparable baseline IL-13 levels were observed in both groups not exposed to DOX. These data demonstrate that MMP-12 is essential for IL-13-driven fibrosis in the lungs. Detailed histological and morphometric analyses are provided in Appendix A. All photomicrographs are presented at 400× original magnification. Data are expressed as mean ± SEM, with *n* = 3 experiments and four mice per group per experiment.

### 3.5. Pharmacological Inhibition of MMP-12 Reduces Allergen-Induced Fibrosis

Given the protective effects observed in MMP-12^−/−^ mice, we next investigated whether MMP-12 inhibitors could prevent fibrosis. We tested two inhibitors, MMP-408 and PF-hydrocholoride (PF-00356231HCl), administered via intraperitoneal injection to mice that had received an intranasal *Af* challenge (Figure 5A). Both inhibitors reduced EPX^+^ eosinophils (Appendix A) and MMP-12^+^ cell accumulation (Figure 5B), collagen deposition (Figure 5C), and the expression of TGF-β (Figure 5D) and α-SMA (Figure 5E) following an *Af* extract challenge compared to vehicle treatment. Our morphometric analysis demonstrated a statistically significant reduction in collagen and all measured profibrotic cytokines (Figure 5B–E, iv). Western blot analysis confirmed these findings, showing a significant reduction in MMP-12, TGF-β, Col IV, and α-SMA in *Af*-challenged inhibitor-treated mice compared to those treated with vehicle (Figure 5F, i,ii). These findings demonstrate that the pharmacological inhibition of MMP-12 can effectively reduce allergen-induced inflammation and fibrosis. The detailed histological analyses of MMP-12, collagen, TGF-β, and α-SMA are provided in Appendix A. All photomicrographs are presented at 400× original magnification, and data are expressed as mean ± SEM, with *n* = 3 experiments and four mice per group per experiment.

### 3.6. Mucus Cell Hyperplasia and Airway Obstruction in Aspergillus fumigatus Extract-Challenged and MMP-12 Inhibitor-Treated Mice

To evaluate the functional outcomes of MMP-12 inhibition, we first examined mucus-producing goblet cell hyperplasia in *Af*-challenged WT mice. PAS staining revealed a substantial decrease in mucus-producing cells in *Af-*challenged WT mice treated with PF-00356231 or MMP-408 compared to vehicle-treated controls (Figure 6A,B). We next examined airway obstruction in *AF*-challenged WT mice treated with vehicle, PF-00356231, or MMP-408 using the Buxco Finepointe RC system. Pulmonary function tests showed improved airway resistance and compliance in *Af-*challenged mice treated with PF-00356231 or MMP-408 compared to vehicle (Figure 6C,D). Lastly, we used ELISA to determine whether MMP-12 inhibitors reduce IL-13 in the lungs of *Af-*challenged mice. Our analysis indicated that both PF-00356231 and MMP-408 treatments significantly lowered IL-13 levels following *Af* exposure. These findings suggest that MMP-12 inhibition downregulates IL-13 to enhance lung function in this mouse model of eosinophilic airway inflammation (Figure 6E). All photomicrographs are presented at 400× original magnification. Data are expressed as mean ± SD, with *n* = 3 experiments and four mice per group per experiment.

## 4. Discussion

MMPs have been widely implicated in airway inflammation and collagen deposition, associated with obstructive pulmonary diseases such as asthma and COPD [19,20,21]. Although matrix protein accumulation is a common feature of many inflammatory diseases, its specific role in driving eosinophilic inflammation-mediated bronchial fibrosis and airway obstruction is still not fully elucidated. Elevated levels of MMP-1, MMP-2, MMP-8, MMP-9, and membrane-type 1 MMP have been detected in the lungs of COPD patients [22,23,24,25]. These enzymes play a pivotal role in the progression of chronic inflammatory conditions, including pulmonary emphysema in COPD. ECM accumulation significantly contributes to COPD pathogenesis, and MMPs actively degrade elastin and other ECM components, thereby impairing lung elasticity. Moreover, MMP-12 degradation products, such as elastin fragments and collagen-derived peptides (e.g., Pro-Gly-Pro), enhance monocyte recruitment and sustain airway inflammation, perpetuating ECM destruction. Notably, MMP-12-deficient mice exposed to cigarette smoke are resistant to emphysema development and show reduced macrophage infiltration, underscoring MMP-12′s central role in disease progression [26,27,28]. While MMP-12 expression is typically low in macrophages under normal conditions, it is markedly increased in alveolar macrophages of smokers with COPD [29]. Elevated levels of MMP-1, MMP-2, MMP-8, and MMP-9 have also been found in the sputum and bronchoalveolar lavage fluid of patients with asthma and COPD [25]. However, the specific contribution of MMP-12 to bronchial fibrosis compared to the other previously reported MMPs in chronic airway diseases remains underexplored. Therefore, this study investigates the role of MMP-12 in regulating bronchial fibrosis and evaluating MMP-12 inhibitors as a potential therapeutic strategy to improve fibrosis-related outcomes and quality of life in patients with chronic airway diseases.

Our study demonstrates that the enzyme MMP-12 is upregulated in allergen-challenged mice and in IL-13-overexpressing mice exhibiting cytokine-driven airway inflammation. This aligns with previous research associating an MMP-12 single nucleotide polymorphism (rs2276109 [−82A→G]) with lung function in individuals with asthma and COPD [29]. Based on this, we hypothesized that MMP-12 induction regulates proinflammatory and profibrotic cytokines that drive EMT and bronchial fibrosis in obstructive airway diseases.

To test this, we used experimental mouse models of allergen- and IL-13-induced airway inflammation. We found that both allergen-challenged MMP-12^−/−^ mice and DOX-exposed, airway-specific MMP-12^−/−^CC10–rtTA-IL-13-overexpressing mice were protected from IL-13, TGF-β, and α-SMA-induced EMT and related fibrosis. This protection was absent in saline-challenged control mice and DOX-exposed CC10–rtTA-IL-13 mice, highlighting the direct role of MMP-12’s profibrotic and proinflammatory functions. Importantly, we observed that DOX-exposed CC10–rtTA-IL-13-overexpressing mice displayed airway pathogenesis resembling that observed in COPD [25]. While TGF-β is known to mediate epithelial, mesenchymal, and immune cell transformation, promoting tissue remodeling and fibrosis [30], several reports suggest that IL-13-induced fibrosis is independent of TGF-β [31,32]. Our evidence shows that bronchial fibrosis driven by either *Aspergillus* exposure or IL-13 overexpression involves MMP-12, which regulates both IL-13 and TGF-β signaling. This suggests that MMP-12 acts upstream of IL-13 to promote TGF-β-mediated fibrotic changes. 

Our mouse models showed that MMP-12 deficiency significantly limits bronchial fibrosis by suppressing IL-13 and TGF-β expression and reducing the accumulation of FSP-1, FGF, and α-SMA. These markers were diminished in allergen-exposed MMP-12^−/−^ mice and DOX-fed MMP-12^−/−^CC10–rtTA-IL-13 mice. Additionally, while IL-13 has been shown to promote inflammatory cell infiltration and airway remodeling, its specific role in airway fibrosis and obstruction has received limited attention [10,33]. Our findings showed that MMP-12 deficiency reduced collagen deposition and the expression of EMT-associated proteins (E-cadherin, N-cadherin, vimentin) and profibrotic cytokines (TGF-β, α-SMA, fibronectin) induced by allergen exposure or IL-13 overexpression. We also showed that MMP-12 deficiency diminished allergen-induced, IL-13-mediated airway resistance and improved lung compliance. These findings emphasize the novel role of MMP-12 in regulating bronchial fibrosis and point to MMP-12 inhibitors as potential therapeutics for human pulmonary disease. Indeed, in the current study, mice challenged with *Aspergillus* and treated with two different MMP-12 inhibitors (PF-00356231HCL and MMP408) showed improved bronchial fibrosis and airway obstruction. These findings are consistent with human studies in which carriers of SNP alleles that lead to reduced MMP12 expression exhibit improved lung function and a lower risk of developing COPD [29]. 

## 5. Conclusions

The evidence presented in this study is both novel and clinically significant for understanding eosinophilic inflammation in asthma pathogenesis. We demonstrated that MMP-12 plays a key regulatory role in eosinophil-responsive cytokines and chemokines, driving EMT and bronchial fibrosis in a mouse model of acute inflammation induced by *Aspergillus fumigatus* and a model of chronic inflammation induced by lung-specific IL-13 overexpression. Notably, treatment with MMP-12 inhibitors protected against bronchial remodeling and fibrosis. MMP-12 also significantly reduced IL-13 and TGF-β expression and inhibited fibroblast and α-SMA-mediated myofibroblast transformation. These findings underscore the therapeutic potential of MMP-12 inhibitors as promising candidates for managing airway remodeling and functional impairment in eosinophilic airway diseases. Based on this evidence, we recommend multicenter, double-blind clinical trials in humans using orally bioavailable MMP-12 inhibitors like PF-00356231HCL and MMP408. Both agents are potent, selective, and ready for clinical use, making them ideal candidates for preventing bronchial remodeling and improving outcomes in patients with eosinophilic inflammation-driven conditions such as chronic rhinosinusitis, asthma, eosinophilic granulomatosis with polyangiitis, and COPD.

## Figures and Tables

**Figure 1 cells-14-01307-f001:**
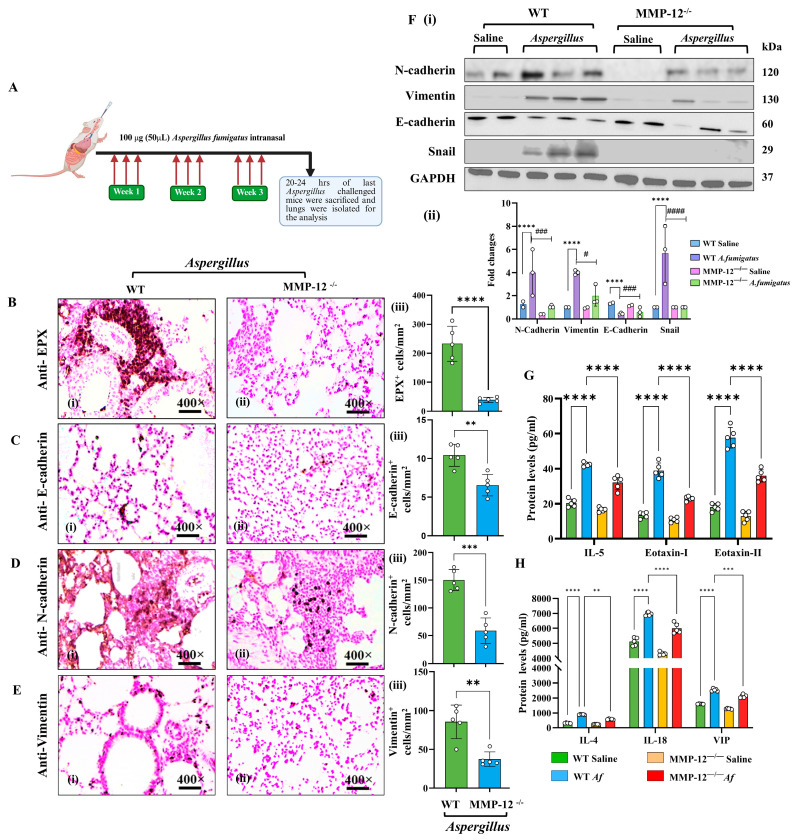
MMP-12-deficient (MMP-12^−/−^) mice accumulate significantly fewer peribronchial and perivascular inflammatory cells in their lungs compared to wild-type (WT) mice following an *Aspergillus fumigatus* (*Af*) extract challenge. Schematic representation of the *Af* extract treatment protocol (**A**). Immunohistochemistry (**B**–**E**, **i**,**ii**) and morphometric analyses (**B**–**E, iii**) show the accumulation of inflammatory cells (anti-EPX) and key factors related to EMT, including collagen, N-cadherin, E-cadherin, and vimentin, in *Af*-challenged WT and MMP-12^−/−^ mice. Western blot (**F**, **i**) and densitometry analysis (**F**, **ii**) were used to measure EMT-associated markers (N-cadherin, E-cadherin, and vimentin), snail, MMP-12, TGF-β, Col I, Col IV, and α-SMA in lung tissues from both WT and MMP-12^−/−^ mice challenged with *Aspergillus* extract. Levels of IL-4, IL-5, IL-18, TGF-β, eotaxin-I, eotaxin-II, and VIP in the serum of *Af*-challenged WT and MMP-12^−/−^ mice were measured by ELISA (**G**,**H**). All data are presented as the mean ± SEM, with *n* = 3 experiments and four mice per group per experiment. ** *p* < 0.01, *** *p* < 0.001, **** *p* < 0.0001, ^#^
*p* < 0.05 vs. WT *Af*, ^###^
*p* < 0.001 vs. WT *Af*, ^####^
*p <* 0.0001 vs WT *Af*. Photomicrographs are 400× original magnification (scale bar, 20 µm).

**Figure 2 cells-14-01307-f002:**
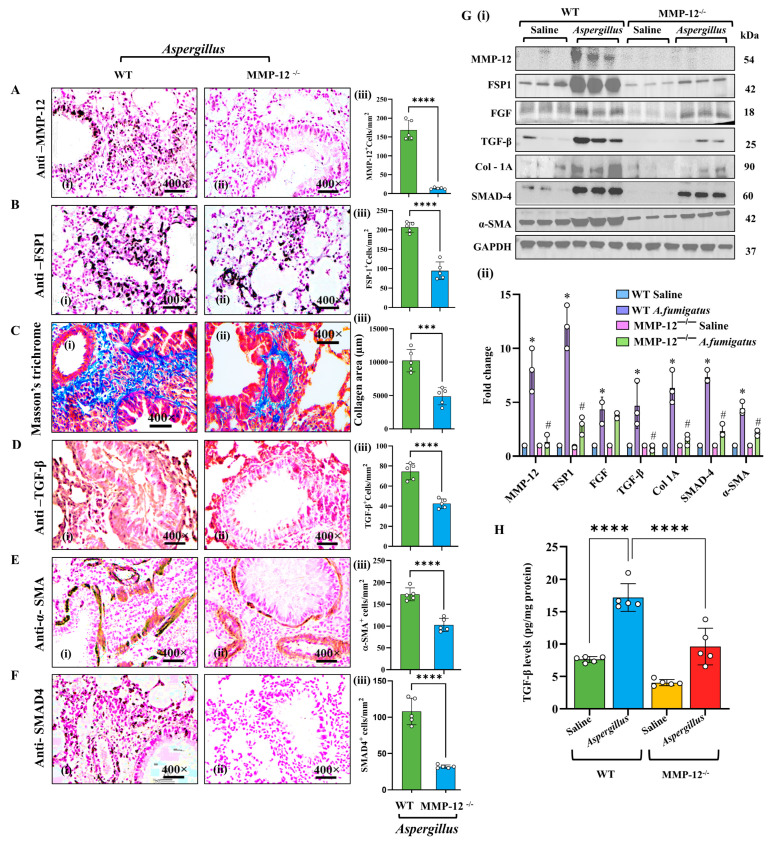
MMP-12 regulates fibrosis-related proteins and genes in *Aspergillus* extract-challenged mice. Immunohistochemistry and morphometric analyses showing levels of fibrosis-associated proteins (MMP-12, FSP-1, TGF-β, α-SMA, and SMAD-4) in WT and MMP12^−/−^ mice challenged with *Aspergillus* (*Af*) extract (**A**–**F**). Western blot and densitometry analysis of fibrosis-related markers (MMP-12, FSP-1, FGF, Col 1A, TGF-β, α-SMA, and SMAD-4) in *Af*-challenged WT and MMP-12^−/−^ lung tissue (**G**, **i**,**ii**). ELISA analysis for TGF-β in saline and *Af*-challenged WT and MMP-12^−/−^ in lung tissue (**H**). Data are expressed as mean ± SEM, with *n* = 3 experiments and four mice per group per experiment. * *p* < 0.05; *** *p* < 0.001, **** *p* < 0.0001, ^#^
*p* < 0.05 vs. WT *Af*. Photomicrographs are presented at 400× original magnification (scale bar, 20 µm).

**Figure 3 cells-14-01307-f003:**
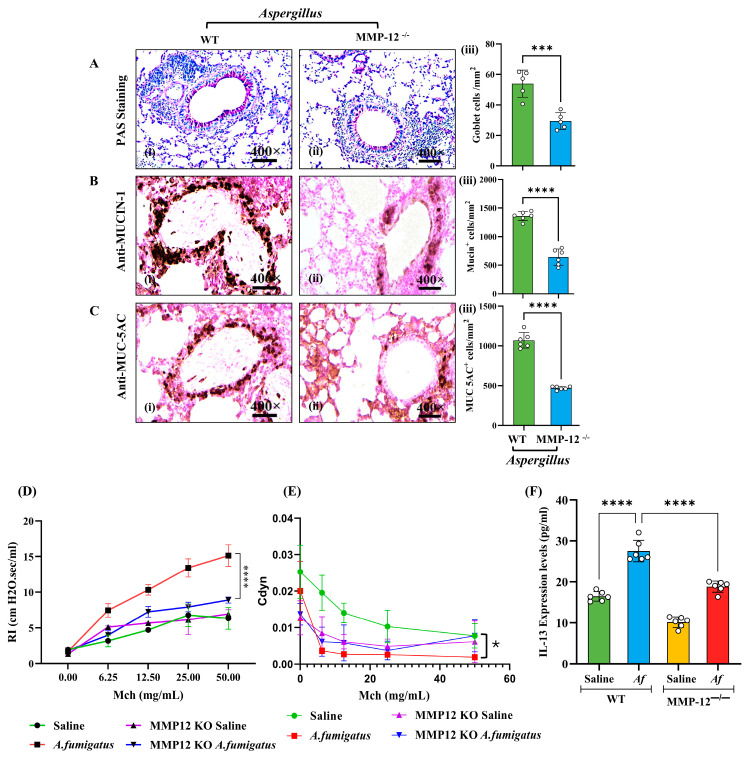
Assessment of lung airway function in *Aspergillus* extract-exposed wild-type and MMP12^−/−^ mice. Representative photomicrographs and quantitation of PAS-positive goblet cells (**A**), MUCIN-1-positive goblet cells (**B**), and MUC5AC-positive goblet cells (**C**) in *Aspergillus fumigatus* (*Af*) extract-challenged MMP-12^−/−^ and wild-type (WT) mice. Airway resistance (RI) and compliance (Cdyn) in response to different concentrations of methacholine (Mch) in *Af*-challenged MMP-12^−/−^ and WT mice (**D**,**E**). Inflammatory cytokine (IL-13) levels were reduced in MMP12^−/−^ mice challenged with *Af* compared to their WT counterparts (**F**). Data are expressed as mean ± SEM, with *n* = 3 experiments and four mice per group per experiment. * *p* < 0.05; *** *p* < 0.001, **** *p* < 0.0001. Photomicrographs are presented at 400× original magnification (scale bar, 20 µm).

**Figure 4 cells-14-01307-f004:**
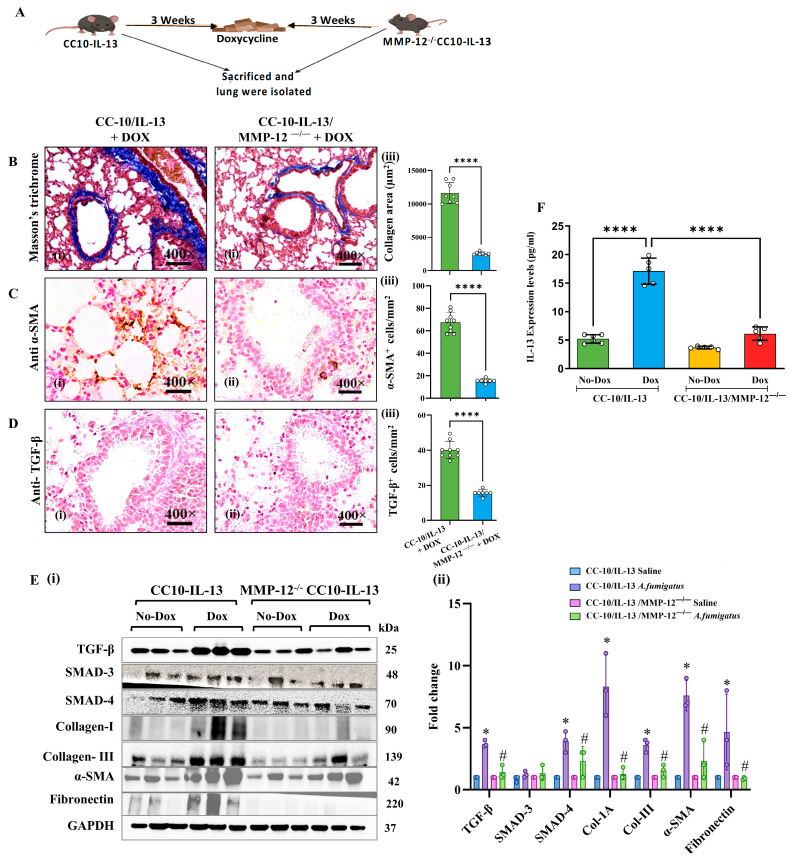
MMP-12 deficiency protects against lung fibrosis in DOX-exposed mice with lung-specific IL-13 overexpression. Schematic showing the DOX exposure protocol (**A**). Representative photomicrographs of lung tissue sections showing collagen accumulation (Masson’s trichrome) and α-SMA and TGF-β expression (immunolabeling) in DOX-exposed CC10–rtTA-IL-13 and MMP-12^−/−^CC10–rtTA-IL-13 mice (**B**–**D**, **i**,**ii**). Quantification of collagen, α-SMA, and TGF-β in these mice (**B**–**D**, **iii**). Western blot and densitometry illustrating the expression of fibrosis-related markers (TGF-β, SMAD-3/SMAD-4, collagen-I/III, α-SMA, and fibronectin) in DOX-exposed CC10/IL-13 mice and DOX-exposed MMP-12^−/−^CC10–rtTA-IL-13 mice (**E**, **i**,**ii**). Densitometric quantitation showed no change in Smad3 (**E**, **ii**). ELISA results depict profibrotic IL-13 expression levels in DOX-exposed CC10–rtTA-IL-13 mice and DOX-exposed MMP-12^−/−^CC10–rtTA-IL-13 mice (**F**). Data are expressed as mean ± SEM, with *n* = 3 experiments and four mice per group per experiment. * *p* < 0.05; **** *p* < 0.0001, ^#^
*p* < 0.05 vs. WT *Af*. Photomicrographs are presented at 400× original magnification (scale bar, 20 µm).

**Figure 5 cells-14-01307-f005:**
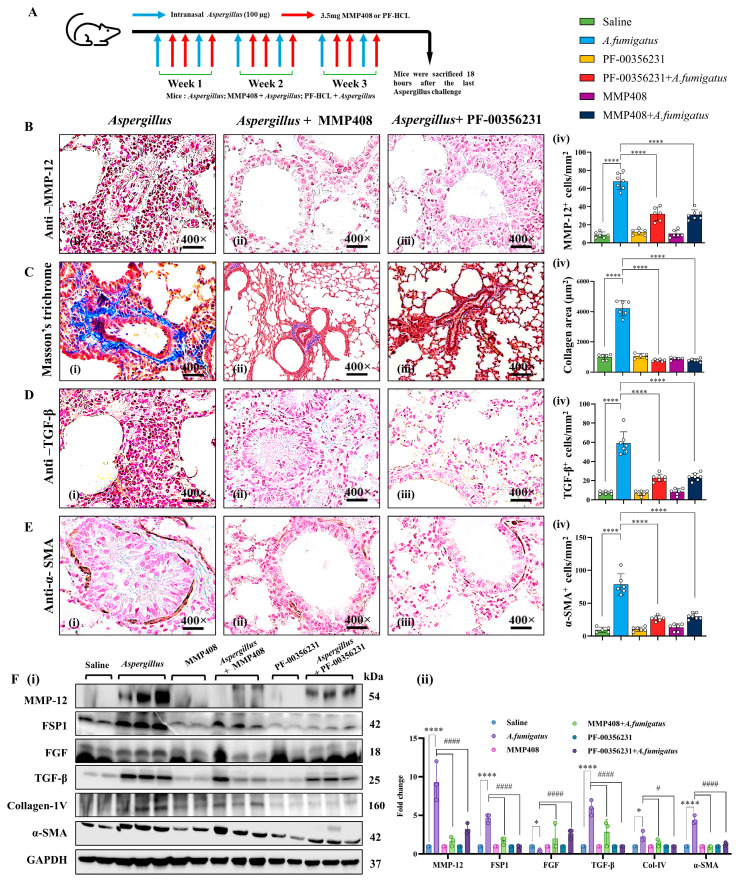
Novel MMP-12 inhibitors (PF-00356231 and MMP408) protect against fibrosis and EMT-associated gene expression in *Aspergillus* (*Af*) extract-challenged WT mice. Schematic representation of the *Af*-challenged mouse model (**A**). Representative immunohistochemical staining and quantification for anti-MMP-12, Masson’s trichrome (collagen), anti-TGF-β, and anti-α-SMA are shown for *Af*-challenged mice treated with vehicle (saline) or MMP-12 inhibitors (**B**–**E**). Western blotting and densitometry for fibrosis-associated markers (TGF-β, α-SMA, Col IV, FSP-1, FGF, and MMP-12) in WT mice treated with vehicle or MMP-12 inhibitors alone and following an *Af* challenge (**F**, **i**,**ii**). Data are expressed as mean ± SEM, with *n* = 3 experiments and four mice per group per experiment. * *p* < 0.05; **** *p* < 0.0001, ^#^
*p* < 0.05 vs. WT *Af*, ^####^
*p* < 0.001 vs. WT *Af*. Photomicrographs are presented at 400× original magnification (scale bar, 20 µm).

**Figure 6 cells-14-01307-f006:**
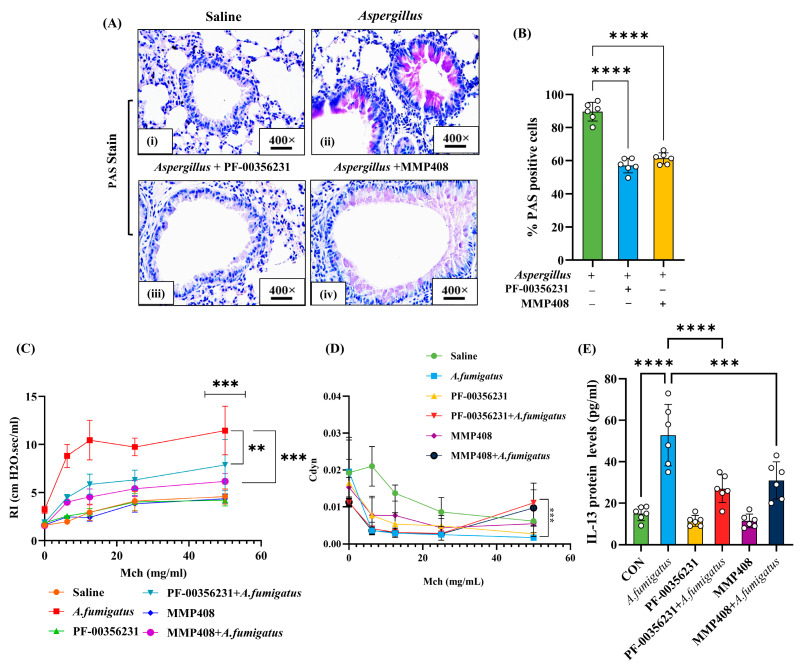
Novel MMP-12 inhibitors (PF-HCL and MMP408) improve eosinophilic inflammation-mediated lung airway dysfunction in *Aspergillus fumigates* (*Af*) extract-challenged mice. Immunohistochemistry staining and quantification of goblet cell hyperplasia in wild-type (WT) mice exposed to saline or challenged with *Af* and treated with MMP-12 inhibitors (**A**,**B**). Airway resistance (RI) and compliance (Cdyn) in response to different concentrations of methacholine (Mch) in *Af*-challenged WT mice, WT mice treated with MMP-12 inhibitors (PF-00356231HCL and MMP408) alone, and *Af*-challenged WT mice treated with inhibitors (**C**,**D**). ELISA data showing profibrotic IL-13 levels in mice challenged with *Af*, mice treated with MMP-12 inhibitors (PF-00356231HCL and MMP408) alone, and mice exposed to *Af* and MMP-12 inhibitors (**E**). Data are expressed as mean ± SEM, with *n* = 3 experiments and four mice per group per experiment. ** *p* < 0.01; *** *p* < 0.001, **** *p* < 0.0001. Photomicrographs are presented at 400× original magnification (scale bar, 20 µm).

## Data Availability

The data presented in this study are available in the article.

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
