# Peer review of "MMP-12 Inhibits Inverse Eosinophilic Inflammation-Mediated Bronchial Fibrosis in Murine Models of Pulmonary Airway Obstruction"

_cells, 2025, doi:10.3390/cells14171307_

Round 1

Reviewer 1 Report

Comments and Suggestions for Authors

General Comments:

This manuscript presents a solid body of experimental data demonstrating the critical role of MMP-12 in eosinophilic inflammation and bronchial fibrosis using Aspergillus-challenged and IL-13-overexpressing mouse models. The combined use of genetic (MMP-12–/– and CC10-IL-13) and pharmacological (PF-HCL, MMP408) approaches strengthens the findings.

However, the manuscript requires major revisions to improve clarity, methodological transparency, and translational relevance. In particular, the discussion should be more balanced, study limitations acknowledged, and some aspects further explored.

Specific Comments:

  1. Originality and Contextualization

    • The authors should better highlight the novelty of their findings relative to prior studies on MMP-12, while avoiding excessive repetition in the discussion.

    • A direct comparison with other MMPs implicated in airway fibrosis (e.g., MMP-9 or MMP-2) would enhance the interpretation.

  2. Methodology

    • The number of animals per group (n=4–6) seems limited for certain histological and functional assessments; a power calculation or mention of biological replicates is needed.

    • It is unclear whether histological and morphometric analyses were conducted blinded. Please clarify to mitigate observer bias.

    • Information on pharmacokinetics and selectivity of the inhibitors should be included or discussed.

  3. Results

    • All IHC images are shown at 400×; lower magnification overviews would help visualize the anatomical context of the pathology.

    • Several analyses are redundant (e.g., WB and IHC for the same proteins) without contributing additional insight.

  4. Discussion

    • Overly enthusiastic statements should be toned down, and a critical discussion of the study’s limitations is needed (e.g., murine model, lack of toxicity data, limited translational information).

    • The translational potential should be discussed more critically, considering real-world challenges such as toxicity, bioavailability, and target selectivity.

  5. Supplementary Material

    • Well-organized and comprehensive. However, streamlining the figures for clarity would improve their utility.

Author Response

Response to Reviewer 1 Responses and Suggestions

General Comments:

This manuscript presents a solid body of experimental data demonstrating the critical role of MMP-12 in eosinophilic inflammation and bronchial fibrosis using Aspergillus-challenged and IL-13-overexpressing mouse models. The combined use of genetic (MMP-12–/– and CC10-IL-13) and pharmacological (PF-HCL, MMP408) approaches strengthens the findings.

We thank reviewer for acknowledging that the manuscript has demonstrated solid body of experimental data to establish the role of MMP-12 in promoting bronchial fibrosis that induce airway obstruction.  The manuscript is thoroughly cheeked, and some plagiarism is found in the methods, which is our limitation.

Additionally, the reviewer raised some concern to improve clarity, methodological transparency, and translational relevance, and study limitations. Accordingly, we revised, and all changes are incorporated in bold letters in the manuscript.

Specific Comments:

  1. Originality and Contextualization
    • The authors should better highlight the novelty of their findings relative to prior studies on MMP-12, while avoiding excessive repetition in the discussion.

Discussion sections revised as suggested.

  • A direct comparison with other MMPs implicated in airway fibrosis (e.g., MMP-9 or MMP-2) would enhance the interpretation.

A comparison of MMP-12 with the earlier reported function of MMP-2 and MMP-9 is incorporated in the discussion section.

  1. Methodology
    • The number of animals per group (n=4–6) seems limited for certain histological and functional assessments; a power calculation or mention of biological replicates is needed.

The number of animals4-6/group/experiments are used; however, we performed (n) = 3 experiments (replicates), which means data of 12-18 mice are used for a power calculation. We mentioned this in each figure.

  • It is unclear whether histological and morphometric analyses were conducted blinded. Please clarify to mitigate observer bias.

All morphometric analysis is formed in blinded manner by two different investigators without knowing the treatments groups.

  • Information on pharmacokinetics and selectivity of the inhibitors should be included or discussed.

MMP408 is specific inhibitor for MMP-12 (PMID: 19278250) and PF-hydrochloride inhibits MMP-12 along with MMP-2 and MMP-9. Both MMP-12 and PF-hcl MMP inhibitors absorbed into the bloodstream after administration. Excretion is the elimination of the drug and its metabolites from the body, primarily through the kidneys. The treatment dose selected based on earlier reported study (PMID: 31242676). The information is now included in the method section.

  1. Results
    • All IHC images are shown at 400×; lower magnification overviews would help visualize the anatomical context of the pathology.

Unfortunately, we took all the images that were taken in 40X magnification.

  • Several analyses are redundant (e.g., WB and IHC for the same proteins) without contributing additional insight.

Immunohistochemistry (IHC) provides cell-specific expression of proteins and quantification of positive cells, whereas western blot analysis validates and offers additional insight into overall protein levels—whether increased or decreased—in the lungs of treated mice with different genetic backgrounds. This information is included in the manuscript.

Discussion

  • Overly enthusiastic statements should be toned down, and a critical discussion of the study’s limitations is needed (e.g., murine model, lack of toxicity data, limited translational information).

We revised the discussion of our results as suggested.

  • The translational potential should be discussed more critically, considering real-world challenges such as toxicity, bioavailability, and target selectivity.

The inhibitors’ toxicity, bioavailability, and target selectivity is now discussed in the manuscript.

  1. Supplementary Material
    • Well-organized and comprehensive. However, streamlining the figures for clarity would improve their utility.

It was difficult to include all histological analyses in the main figure. Therefore, to enhance clarity and benefit the readers, we have provided photomicrographs of all treatment groups; quantified for statistical comparison of positive cells in the supplementary figures.

  1. Conclusion of the study is provided.

Thanks.

Sincerely,

Anil Mishra, PhD

Professor of Medicine

Reviewer 2 Report

Comments and Suggestions for Authors

To the Editor,

The Authors investigated an interesting topic of the contribution of matrix metalloproteinase-12 (MMP12) in the propagation of airway inflammation and remodeling. They used wild type (WT) or Mmp12-/- (knock-out of the Mmp12 gene) mice sensitized intranasally with Aspergillus fumigatus (ASP) or crossbred with mice showing lung-targeted expression of IL-13 (CC10-rtTA-IL-13). In a series of interesting experiments, they confirmed, among other findings, an important role of MMP12 in IL-13-driven inflammation and lung eosinophilia. The study is concise and should be of interest to researchers in the field of airway immunology. I don’t have many major concerns with this study, except for a plea to improve data presentation and correct numerous writing errors. These and other, rather technical comments are listed below to be considered by the Editor and Authors.   

Major

  1. Figures and legends need to be checked for errors. I list some of them in the minor comments section below. To improve clarity, I suggest changing the gray/pattern design in the bars (black data points are not always clearly visible) with adding plain colors. To improve reading of the Figure legends and data tracking, the letters referring to the subsequent parts of the Figures should precede the descriptions.
  2. I am not a native English speaker, but I think the text should be thoroughly checked for typos, grammar errors, and style. Abstract should also be shortened and clarified, for example could you somehow remove DOX-exposed and DOX-food exposed, as it is was rather a standard procedure to trigger lung IL-13 overexpression.

Minor

  1. The beginning of the Introduction section needs to be rephrased. ‘Eosinophil-associated airway inflammation’ refers to the presence of eosinophils in the airway mucosa, but this cannot be considered a distinct disease (line 34), as there are many conditions with variable extent of airway eosinophilia (CRS, asthma, EGPA, even COPD). Similarly, eosinophilia is not a defining feature of asthma or CRS.
  2. The study Ref. 12 confirmed that IL-13 overexpression in the murine lungs led to increased expression of MMPs and cysteine proteases. They did not confirm ‘matrix metalloproteinases (MMPs) as a potent inducer of IL-13 induced eosinophilic inflammation’, as suggested in the Introduction (lines 47-48). Please verify this and update the citations.
  3. The sentence in lines 53-56 is very unclear.
  4. The line 60-62 describing the model as ‘airway-specific MMP-12-deficient-CC10-IL-13 overexpressed mice that have deficiency of MMP-12 specifically in the airways‘ is awkward and not quite correct. Based on the results presented, these were mice with lung(airway)-specific overexpression of IL-13 (Tet-On) either on a normal or Mmp12-/- background. In other words, Mmp12 deficiency (in Mmp12-/- mice) was not restricted to the airways. Perhaps more details on strains and generation of bitransgenic mice are necessary. Currently, it is described briefly in section 2.1, without methodological details or references to the team's previous publications where this model was developed. The Methods section and supplements should be detailed enough to allow other teams to replicate all the experiments.
  5. The description of bitransgenic mice should be standardized throughout the text, I prefer CC10-rtTA-IL-13, as it refers to the original study (PMID:11067861) and the design of the constructs (CC10-rtTA plus tetO-IL-13).
  6. The Authors should explain the reasons for using an antigen exposure model with intranasal instillation of Aspergillus fumigatus extract that was not followed by aerosol inhalation challenge, which could be more reasonable for a lung study. I think they might have simplified the model because they observed pathological changes in the lungs already after nasal administration of the extracts. Perhaps this is also preferable in experiments with lung induced expression of mediator promoting inflammatory response (IL-13 in this case). I am not an expert on the animal allergen exposure model; this is just to explain the rationale behind this protocol to the reader.
  7. Line 159. A hallmark of EMT is a switch in expression of E- and N-cadherins, E-cadherin is negatively regulated and should not be put in line with renown, say positive, EMT markers.
  8. Please thoroughly check the assignment of statistics symbols on the graphs. Some obvious significant differences are not marked, eg, Fig. 1GH, in Mmp12-/- datasets, the difference between ‘saline’ and ‘ASP’ is not assigned significant?, even for Eotaxin-2 or IL-18? I understand that the Authors could have removed these markings for clarity; however, this is inconsistent as in some places ‘saline vs ASP’ datasets are assigned (eg, for most of WT datasets). Please check all Figures, anyway.
  9. The influence of Mmp12 ko. on ASP-induced Snail expression is spectacular (Fig. 1F)! In essence, it is turned off. Perhaps, the Authors have additional IHC data that could complement this very interesting result.
  10. I understand that WB was a complementary method. However, the Authors may consider showing normalized WB band densities (eg, in the supplement), which could better reflect the results discussed in the text. For example, it is hardly seen in Fig. 1F that cadherin-E was increased in Mmp12-/- ASP compared to WT ASP, as suggested in the text (line 165), perhaps densities could help here. This is just my suggestion, not necessarily to be introduced by the Authors.
  11. There is no legend for Figure 2H, is this TGF measured in blood samples or lung tissue homogenate.
  12. Section 3.3 and Fig. 3F. In what kind of samples was IL-13 measured, blood or lung.
  13. The information in lines 173-175 (n, replicates) and similar should be moved to the Figure legends.
  14. The labels in Fig. 4B-D [iii] ‘double’ and ‘triple’ are not clear, they should be rather Mmp12+/+ and Mmp12-/-. Please, check. Perhaps introducing the term Mmp12+/+ would help track the results more easily in section 3.4 and in the discussion.
  15. Please check the labels in ‘e’ panels of suppl. Fig. 4. Aren’t Dex and no-Dox swapped in Mmp12-/- dataset?
  16. Please update the description of the model in lines 300-301. I assume these were WT mice challenged with ASP.
  17. I suggest using PF-00356231 instead of PF-HCL in text and graphs.
  18. Please add references in line 391 (eg, PMID 20018959).
  19. Line 402, Mmp12-/- mice were protected against an enhanced inflammatory response compared to WT (Mmp12+/+) but not to the ‘saline challenged control-mice’ as it was put there. This was due to the absence of additional profibrotic and proinflammatory function of MMP12 in Mmp12-/-.
  20. Line 405. ‘… observed in COPD patients.’ Please add a reference.
  21. Line 407. MMP12 does not limit fibrosis (?). Perhaps it should be ‘MMP12 deficiency’.
  22. Please explain some additional abbreviations, such as FSP1, DOX etc.
  23. Thinking about any potential applications of the results, I am just curious if Mmp12 deficiency (or inhibition) could have any adverse effects on lung biology. For example, there are reports suggesting the contribution of MMP12 to resolution of inflammation. Did the Authors notice any baseline changes in lung development or function in Mmp12-/- mice compared to WT. In humans, carriers of SNP alleles associated with reduced expression of MMP12 are characterized by better lung function and decreased risk of COPD (PMID 20018959).

Author Response

Response to Reviewer 2 Concerns and Suggestions

We thank reviewer for acknowledging that the topic is interesting on the contribution of matrix metalloproteinase-12 (MMP12) in the propagation of airway inflammation and remodeling. The study is concise and should be of interest to researchers in the field of airway immunology. I don’t have many major concerns with this study, except for a plea to improve data presentation and correct numerous writing errors. The author’s point-by-point responses to the reviewer’s suggestion and technical comments are as follows

Major

  1. Figures and legends need to be checked for errors. I list some of them in the minor comments section below. To improve clarity, I suggest changing the gray/pattern design in the bars (black data points are not always clearly visible) with adding plain colors. To improve reading of the Figure legends and data tracking, the letters referring to the subsequent parts of the Figures should precede the descriptions.

It has been modified in the revised manuscript

  1. I am not a native English speaker, but I think the text should be thoroughly checked for typos, grammar errors, and style. Abstract should also be shortened and clarified, for example could you somehow remove DOX-exposed, and DOX-food exposed, as it was rather a standard procedure to trigger lung IL-13 overexpression.

As per the reviewer’s suggestion, we removed “DOX-exposed and DOX-food exposed” from the abstract.

Minor

  1. The beginning of the Introduction section needs to be rephrased. ‘Eosinophil-associated airway inflammation’ refers to the presence of eosinophils in the airway mucosa, but this cannot be considered a distinct disease (line 34), as there are many conditions with variable extent of airway eosinophilia (CRS, asthma, EGPA, even COPD). Similarly, eosinophilia is not a defining feature of asthma or CRS.

We agree with the reviewer and replace “eosinophils” to inflammatory cells.

  1. The study Ref. 12 confirmed that IL-13 overexpression in the murine lungs led to increased expression of MMPs and cysteine proteases. They did not confirm ‘matrix metalloproteinases (MMPs) as a potent inducer of IL-13 induced eosinophilic inflammation’, as suggested in the Introduction (lines 47-48). Please verify this and update the citations.

We apologies and corrected the sentence on line 47-48 and provided other citation for remodeling and fibrosis.

  1. The sentence in lines 53-56 is very unclear.

Clarity to the sentence is provided and change is highlighted.

  1. The line 60-62 describing the model as ‘airway-specific MMP-12-deficient-CC10-IL-13 overexpressed mice that have deficiency of MMP-12 specifically in the airways‘is awkward and not quite correct. Based on the results presented, these were mice with lung(airway)-specific overexpression of IL-13 (Tet-On) either on a normal or Mmp12-/- background. In other words, Mmp12 deficiency (in Mmp12-/- mice) was not restricted to the airways. Perhaps more details on strains and generation of bitransgenic mice are necessary. Currently, it is described briefly in section 2.1, without methodological details or references to the team's previous publications where this model was developed. The Methods section and supplements should be detailed enough to allow other teams to replicate all the experiments.

As suggested the sentence is modified by mentioning “lung(airway)-specific overexpression of CC10-rtTA-IL-13 bitransgenic mice and MMP12-/- background mice.”  The MMP-12 deficient- CC10-IL-13 overexpressed mice were generated in our laboratory

  1. The description of bitransgenic mice should be standardized throughout the text, I prefer CC10-rtTA-IL-13, as it refers to the original study (PMID:11067861) and the design of the constructs (CC10-rtTA plus tetO-IL-13).

We agree with the reviewer that these mice are generated by Dr.Elias and included his reference to our manuscript. We corrected as suggested by the reviewer. Now, we better defined the details of MMP-12-/-CC10-rtTA-IL-13 mice generation in 2.1. Further, we mentioned CC10-rtTA-IL-13 throughout the text.

  1. The Authors should explain the reasons for using an antigen exposure model with intranasal instillation of Aspergillus fumigatus extract that was not followed by aerosol inhalation challenge, which could be more reasonable for a lung study. I think they might have simplified the model because they observed pathological changes in the lungs already after nasal administration of the extracts. Perhaps this is also preferable in experiments with lung induced expression of mediator promoting inflammatory response (IL-13 in this case). I am not an expert on the animal allergen exposure model; this is just to explain the rationale behind this protocol to the reader.

We agree with the reviewer that several investigators used aerosol inhalation. However, we prefer intranasal Aspergillus challenged model, which was developed by us and published in 2001 J. Clin Investigation, 107: 83-90. (PMCID: PMC198543) and since then, we published < 30 manuscript using the same model. We provided this reference in the manuscript for the readers connivence.

  1. Line 159. A hallmark of EMT is a switch in expression of E- and N-cadherins, E-cadherin is negatively regulated and should not be put in line with renown, say positive, EMT markers.

We apologies and excluded E-cadherin, now from line 174.

  1. Please thoroughly check the assignment of statistics symbols on the graphs. Some obvious significant differences are not marked, eg, Fig. 1GH, in Mmp12-/- datasets, the difference between ‘saline’ and ‘ASP’ is not assigned significant?, even for Eotaxin-2 or IL-18? I understand that the Authors could have removed these markings for clarity; however, this is inconsistent as in some places ‘saline vs ASP’ datasets are assigned (eg, for most of WT datasets). Please check all Figures, anyway.

The statistics symbols were incorporated in the revised manuscript

  1. The influence of Mmp12 ko. on ASP-induced Snail expression is spectacular (Fig. 1F)! In essence, it is turned off. Perhaps, the Authors have additional IHC data that could complement this very interesting result.

We tried to verify this Snail data, but the antibody did not work for IHC; I hope reviewer will understand that this is our limitation.

  1. I understand that WB was a complementary method. However, the Authors may consider showing normalized WB band densities (eg, in the supplement), which could better reflect the results discussed in the text. For example, it is hardly seen in Fig. 1F that cadherin-E was increased in Mmp12-/- ASP compared to WT ASP, as suggested in the text (line 165), perhaps densities could help here. This is just my suggestion, not necessarily to be introduced by the Authors.

WB densitometry was included in the updated manuscript

  1. There is no legend for Figure 2H, is this TGF measured in blood samples or lung tissue homogenate.

We added the legend of figure 2H

  1. Section 3.3 and Fig. 3F. In what kind of samples was IL-13 measured, blood or lung.

We added the revised manuscript

  1. The information in lines 173-175 (n, replicates) and similar should be moved to Figure legends.

We now mentioned same in the figure 1 legends.

  1. The labels in Fig. 4B-D [iii] ‘double’ and ‘triple’ are not clear, they should be rather Mmp12+/+ and Mmp12-/-. Please, check. Perhaps introducing the term Mmp12+/+ would help track the results more easily in section 3.4 and in the discussion.

We followed the similar name throughout the revised manuscript

  1. Please check the labels in ‘e’ panels of suppl. Fig. 4. Aren’t Dex and no-Dox swapped in Mmp12-/- dataset?

We thank, it was swapeped in DOX vs no DOX MMP-12-/- mice.

  1. Please update the description of the model in lines 300-301. I assume these were WT mice

challenged with ASP.

  1. I suggest using PF-00356231 instead of PF-HCL in text and graphs.

Added in the revised manuscript

  1. Please add references in line 391 (eg, PMID 20018959).

Added the reference

  1. Line 402, Mmp12-/- mice were protected against an enhanced inflammatory response compared to WT (Mmp12+/+) but not to the ‘saline challenged control-mice’ as it was put there. This was due to the absence of additional profibrotic and proinflammatory function of MMP12 in Mmp12-/-.

  1. Line 405. ‘… observed in COPD patients.’ Please add a reference.

Reference provided, now on line 414

  1. Line 407. MMP12 does not limit fibrosis (?). Perhaps it should be ‘MMP12 deficiency’.

Corrected as suggested, now on line 416.

  1. Please explain some additional abbreviations, such as FSP1, DOX etc.

We provided abbreviation of DOX in section 2.1, and FSP1 in section 2.5.

  1. Thinking about any potential applications of the results, I am just curious if Mmp12 deficiency (or inhibition) could have any adverse effects on lung biology. For example, there are reports suggesting the contribution of MMP12 to resolution of inflammation. Did the Authors notice any baseline changes in lung development or function in Mmp12-/- mice compared to WT. In humans, carriers of SNP alleles associated with reduced expression of MMP12 are characterized by better lung function and decreased risk of COPD (PMID 20018959).

Studies in MMP-12-deficient mice have primarily focused on their resistance to emphysema development and their involvement in lung fibrosis models, but there's limited information regarding specific lung development abnormalities in MMP12-/- mice, the lung development studies are very different than the functional abnormalities developed in adult mice. Regarding functional abnormalities, we did not notice any problems in lung function, e.g. resistance and compliance experiments between naïve WT and MMP-12-/- mice, as shown in Figure 3 D & E. However, the data indicates decreased Th2 cytokines in MMP-12-/-naïve mice compared to WT naïve mice (Figure 3F), indicating MMP-12 may cause some defect in T cells function, but that needs another study to establish.

Thanks.

Sincerely,

Anil Mishra, PhD

Professor of Medicine

Reviewer 3 Report

Comments and Suggestions for Authors

Study by Kathera et al. employs elegant animal models and various histological techniques to deduce the role of MMP12 in eosinophilic inflammation mediated bronchial fibrosis. While the findings are intriguing, there are several aspects that require further clarification and discussion. Addressing these points would enhance the clarity and overall impact of the study.

Major Comments

  1. The authors demonstrate that Aspergillus-challenged MMP12 KO mice exhibit reduced eosinophilic infiltration and EMT marker expression. However, these effects are not explored in subsequent animal studies utilizing MMP12 inhibitors. It would significantly strengthen the study to show whether pharmacologic inhibition of MMP12 can recapitulate the phenotypes observed in KO models.
  2. Figure 4F indicates a reduction in IL-13 protein levels even in CCL10/IL-13 overexpressing mice on an MMP12 KO background. The mechanistic basis for this observation remains unclear and should be addressed. Potential explanations should be discussed.
  3. It would be informative to determine whether MMP12 expression is elevated in Dox-treated CCL10/IL-13 mice compared to untreated controls. This would clarify whether MMP12 acts downstream of IL-13 signaling in promoting fibrotic changes, providing a rationale for the observed protective effects of MMP12 deficiency.
  4. Does doxycycline-induced IL-13 expression in CCL10/IL-13 mice lead to goblet cell hyperplasia? If so, is this phenotype attenuated in MMP12 KO mice? Including this analysis would offer a more comprehensive view of the role of MMP12.
  5. Please provide the statistical analysis for Figure 3D and 3E. Further, it doesn’t seem that MMP12 KO in Aspergillus challenged mice, leads to an improvement in compliance. It would be important to see the statistical analysis to conclude anything.
  6. Line 334 states that compliance is improved in MMP12 inhibitor-treated mice. However, the relevant data are not shown. Please include the raw data and statistical analysis to support this claim.
  7. Please provide airway resistance data for Figure 4.
  8. The cellular source of MMP12 and its regulation remain unclear. Is MMP12 produced by epithelial cells, macrophages, or another cell type? Does IL-13 directly regulate MMP12 expression or activity? Conversely, could MMP12 influence IL-13 production or secretion? These mechanisms should be more thoroughly discussed in the Discussion section.
  9. The changes in Smad3 and Smad4 expression in Figure 4E are not visually apparent. Please provide higher-resolution images and include quantitative densitometric analysis to support the stated conclusions.
  10. The authors analyze different collagen isoforms in various figures—Collagen IV and Collagen I in Figure 2, Collagen I and Collagen III in Figure 4, and Collagen IV in Figure 5. However, the rationale for selecting these specific isoforms at different stages or in different models is not clearly explained. It would strengthen the manuscript to provide a brief justification for this choice.

Minor comments

  1. 1. Line 276 refers to Supplementary Figure 4, which is not included in the manuscript. Please revise the text accordingly.
  2. Line 302, the sentence is incomplete.
  3. The dark colors used in the bar graphs obscure visibility of individual data points. Please revise the color scheme to improve readability.
  4. Please provide reference supporting the statement in line 244 in the results section.
  5. The sentence in line 262 is grammatically wrong.
Comments on the Quality of English Language

The manuscript would benefit from careful proofreading to correct grammatical errors. Several sections could be improved with more concise and precise language.

Author Response

Responses to the Reviewer 3 Concerns and Suggestions

We that reviewer #3 for acknowledging that finding reported are intriguing; but need more clarification and discussion to enhance the clarity and overall impact of the study. Please find below the point-by-point responses to the reviewer’s comments and included all suggestion in the manuscript.

Major Comments

  1. The authors demonstrate that Aspergillus-challenged MMP12 KO mice exhibit reduced eosinophilic infiltration and EMT marker expression. However, these effects are not explored in subsequent animal studies utilizing MMP12 inhibitors. It would significantly strengthen the study to show whether pharmacologic inhibition of MMP12 can recapitulate the phenotypes observed in KO models.

We acknowledge the reviewer suggestion to examine all parameters examined in MMP-12 gene-deficient mice should also be explored in MMP-12 inhibitors treated mice. We included the anti-EPX+ eosinophils in the Supplementary figure 5 E. However, the aim of our study is to understand the role of MMP-12 in promoting bronchial fibrosis and lung function; therefore, we focused MMP-12 inhibitors study limited to evaluating eosinophilic inflammation mediated bronchial fibrosis and associated airway function. Accordingly, we only examined fibrosis associated protein, collagen accumulation (not EMT proteins) to propose that the MMP-12 inhibitors are the novel candidate to treat lung fibrosis related airway obstruction.  Hope reviewer will understand or limitation to examine each parameter as reported in gene-deficient mice.

  1. Figure 4F indicates a reduction in IL-13 protein levels even in CCL10/IL-13 overexpressing mice on an MMP12 KO background. The mechanistic basis for this observation remains unclear and should be addressed. Potential explanations should be discussed.

The IL-13 is a profibrotic cytokine and our presented data show that MMP-12 regulate IL-13 in Aspergillus challenged mouse model of asthma (Figure. 3 F). Therefore, the reduced IL-13 observed in MMP12 KO background CC10-IL-13 mice validated the fact that MMP-12 indeed regulate IL-13 induced fibrosis associated airway obstruction and we discussed this in the manuscript line number 425-428.

  1. It would be informative to determine whether MMP12 expression is elevated in Dox-treated CCL10/IL-13 mice compared to untreated controls. This would clarify whether MMP12 acts downstream of IL-13 signaling in promoting fibrotic changes, providing a rationale for the observed protective effects of MMP12 deficiency.

The presented data indicates that MMP12 acts upstream of IL-13 signaling in promoting fibrotic changes, because DOX regulated MMP-12-/-CC10-IL-13 mice down regulate IL-13 and TGF-β compared to the DOX exposed CC10-IL-13 mice, mentioned on the line number 428-429.

  1. Does doxycycline-induced IL-13 expression in CCL10/IL-13 mice lead to goblet cell hyperplasia? If so, is this phenotype attenuated in MMP12 KO mice? Including this analysis would offer a more comprehensive view of the role of MMP12.

Yes, 3-4 weeks doxycycline exposed IL-13 expression in CCL10/IL-13 mice develop goblet cell hyperplasia; however, these develop chronic inflammation with highly induced accumulation of collagen and mucus, and difficult to have good, focused photomicrograph of goblet cells to provide in the manuscript.

  1. Please provide the statistical analysis for Figure 3D and 3E. Further, it doesn’t seem that MMP12 KO in Aspergillus challenged mice, leads to an improvement in compliance. It would be important to see the statistical analysis to conclude anything.

We apologies for missing the statistical significance, Added in the revised manuscript

  1. Line 334 states that compliance is improved in MMP12 inhibitor-treated mice. However, the relevant data are not shown. Please include the raw data and statistical analysis to support this claim.

The data is provided in figure 6.

  1. Please provide airway resistance data for Figure 4.

Difficult to provide airway resistance or compliance data in 3-4 weeks doxycycline exposed CCL10/IL-13 mice, because due to chronic inflammation these mice do not survive in response to different concentrations of methacholine (Mch) challenge.

  1. The cellular source of MMP12 and its regulation remain unclear. Is MMP12 produced by epithelial cells, macrophages, or another cell type? Does IL-13 directly regulate MMP12 expression or activity? Conversely, could MMP12 influence IL-13 production or secretion? These mechanisms should be more thoroughly discussed in the Discussion section.

Our analysis indicates that macrophages are the source of MMP-12 in the lungs of aspergillus challenged mice showed in supplementary figure 6.

  1. The changes in Smad3 and Smad4 expression in Figure 4E are not visually apparent. Please provide higher-resolution images and include quantitative densitometric analysis to support the stated conclusions.

We provided high resolution corrected Figure with quantitative densitometric analysis.

  1. The authors analyze different collagen isoforms in various figures—Collagen IV and Collagen I in Figure 2, Collagen I and Collagen III in Figure 4, and Collagen IV in Figure 5. However, the rationale for selecting these specific isoforms at different stages or in different models is not clearly explained. It would strengthen the manuscript to provide a brief justification for this choice.

Please note, we provided Collagen I, Collagen IV in Figure 1; Collagen-1 in Figure 2, as Collagen IV was not detectable (either degraded or technical problem), and Collagen IV in Figure 5, Collagen 1 (either degraded or technical problem)

Minor comments

  1. Line 276 refers to Supplementary Figure 4, which is not included in the manuscript. Please revise the text accordingly.

Supplementary Figure 4, included in the manuscript.

  1. Line 302, the sentence is incomplete.

Line 302 (now 324), sentence is now completed.

  1. data points. Please revise the color scheme to improve readability.

Data points are revised

  1. Please provide reference supporting the statement in line 244 in the results section.

Reference provided now in line 264.

  1. The sentence in line 262 is grammatically wrong.

The sentence line now 283 is corrected

The manuscript would benefit from careful proofreading to correct grammatical errors. Several sections could be improved with more concise and precise language.

We carefully proofread and corrected all grammatical errors by the experts and highlighted the corrected sentences.

Thanks.

Sincerely,

Anil Mishra, PhD

Professor of Medicine

Reviewer 4 Report

Comments and Suggestions for Authors

In this study, the authors present that MMP-12 contributes to airway inflammation and fibrosis in mice exposed to allergens or over-expressing IL-13. According to their findings genetic deletion or pharmacological inhibition of MMP-12 improves lung function, reduces fibrosis markers, and lowers pro-inflammatory cytokine levels.

The manuscript is well-written, however some issues require further clarification.

Comments/Suggestions:

1. How the mice findings can be translated to human asthma/COPD ?

2. Did you explore any gender (male vs. female mice) differences in the response ?

3. The sample size is limited. Only 3-4 mice are per group, so this can the statistical power of the analyses and increase the risk for any false negative findings. 

4. In the Methods section, all statistical techniques mentioned are parametric. Did you check for the normality of distributions ? 
If you have done this, please describe how it was done in the Methods section. Otherwise, you need to perform this (e.g., by applying Shapiro-Wilk test) for each group (not to all data pooled together). If deviation of normality is found, you should re-perform the entire statistical analysis by using non-parametric methods.

5. In line with previous comment, where you applied ANOVA you should further check for the similarity of variance among the compared groups. 

6. Did you test any organs in order to ensure that these inhibitors did not lead to side effects elsewhere?

Author Response

Responses to Reviewer 4 comments and suggestions

We thank reviewer for, acknowledging that the manuscript is well-written that shows MMP-12 contribution in promoting airway inflammation and fibrosis following allergens exposure or over-expressing IL-13. The presented findings of MMP-12 genetic deletion or pharmacological inhibition improve lung function, reduce fibrosis markers, and lower pro-inflammatory cytokine levels. However, some issues raised that we are responding in point-by-point below. 

Comments/Suggestions:

  1. How can the mice findings be translated to human asthma/COPD?

The induced MMP-12 is already reported in human asthma and COPD. Our study establishes the critical role of induced MMP-12 in promoting asthma and COPD pathogenesis.  On line 451-455, we reported studies that showed “carriers of SNP alleles associated with reduced MMP12 expression exhibit improved lung function and a lower risk of developing COPD.”  Accordingly, the translation of these findings will be using MMP-12 inhibitors as promising therapeutic candidates to treat bronchial fibrosis and airway obstruction for human asthma and COPD.

  1. Did you explore any gender (male vs. female mice) differences in the response?

We used both male and female mice in our experiments and stated in section 2.1, we did not find any markable differences regarding MMP-12 induction or reversal by genetic deletion or pharmacological inhibition.

  1. The sample size is limited. Only 3-4 mice are per group, so this can the statistical power of the analyses and increase the risk for any false negative findings. 

The sample size is 3-4 mice/group/experiments and n=3 experiment. Thus, for statistical power analysis we used total 9-12 mice per group. Which is listed in each result section.

  1. In the Methods section, all statistical techniques mentioned are parametric. Did you check for the normality of distributions? 

If you have done this, please describe how it was done in the Methods section. Otherwise, you need to perform this (e.g., by applying Shapiro-Wilk test) for each group (not to all data pooled together). If deviation of normality is found, you should re-perform the entire statistical analysis by using non-parametric methods.

Please note that p-value of < 0.05 was considered statistically significant using non-parametric methods, and stated in section 2.9.

  1. In line with the previous comment, where you applied ANOVA, you should further check for the similarity of variance among the compared groups. 

   We used an F-test to compare the variances of the two samples.

  1. Did you test any organs in order to ensure that these inhibitors did not lead to side effects elsewhere?

During Autopsy of the mice, we did not observe any visible alterations in any experimental mice compared to the control mice.

We hope that all reviewers should appreciate our responses to each quarry and suggestion provided in this letter of response and included in the manuscript. Still if any reviewer has any other query or suggestion to improve the quality of the study, please feel free to contact again to improve the quality of the manuscript.

Thanks.

Sincerely,

Anil Mishra, PhD

Professor of Medicine

Round 2

Reviewer 2 Report

Comments and Suggestions for Authors

To the Editor,

The Authors have addressed most of my technical suggestions. Most notably, Western blot densitometry with statistical analysis has been added, and the nomenclature has been partially standardized. Although the Authors chose not to modify the appearance of the bars in the main graphs (I had suggested using plain colors without patterns to improve the visibility of data points), I accept this decision. The method of marking statistical differences on the graphs using symbols still appears inconsistent in some places. Fortunately, the data speak for themselves—differences between control and Af (or IL-13 induction) groups are clearly evident in most experiments. Overall, this is a very interesting study. I would like to congratulate Professor Anil Mishra and his team on their work.

Minor

  1. The Authors still claim that in their 'Mmp12-/- crossed with the CC10-rtTA-IL-13 mice' the deficiency of MMP12 was restricted to the lungs, which is not the case (eg, line 61 which reads ‘MMP12 deficiency specifically in the lung airways’). Perhaps the problem lies in complicated re-writing in lines 60-62.
  2. Again, I plead for careful and thorough review of the manuscript to correct writing errors throughout. Just a few examples: Line 153. It should be ‘PF-00356231-hydrochloride (HCl)’ instead of ‘PF-hydrochloride (hcl)’. Line 157, yes this is information is provided, no need to write this down. Figures: the Authors changed the ASP abbreviation to Af in the main text, but ASP is still retained in some Figures (eg, Fig. 1h, 3f..). To unify nomenclature, please change the label ‘PF-HCl’ to ‘PF-00356231’ in the blot shown in Fig. 5F(i). etc.
  3. Conclusions: Why these proposed studies should focus specifically on ‘improving quality of life’ in eosinophilic inflammatory diseases (line 523)? I think a simpler ‘disease management’ or ‘improvement in clinical outcomes’ could be better.

Author Response

Response to Reviewer 2 Responses and Suggestions

We addressed all reviewers’s queries and incorporated all suggestions in the revised manuscript.

I had suggested using plain colors without patterns to improve the visibility of data points.

Response: In the revised manuscript, we made all the bar graphs in plain color so that readers could visualize the data points.

Minor Suggestions:

  1. We rewrote the sentence in lines 60-62.
  2. To unify the nomenclature, we corrected ‘PF-HCl’ to ‘PF-00356231’ in the blot shown in Fig. 5F(i), etc., and the manuscript text.
  3. We changed the ASP abbreviation to Af in the main text.
  4. In conclusion, as suggested, we corrected the sentence “improvement in clinical outcomes” in place of “improving quality of life.”

Reviewer 3 Report

Comments and Suggestions for Authors

For comment 8, authors mention that the results are provided in supplementary figure 6. However, there is no supplementary Figure 6 provided in the files.

In response to comment 9 , authors provided densitometric quantifications for Smad3 and Smad 4. As Smad 3 doesnot show any differences between the treatments, it should be mentioned accordingly in the results section (Line 322-323). 

Author Response

For comment 8, the authors mention that the results are provided in supplementary figure 6. However, there is no supplementary Figure 6 provided in the files.

Response: Due to a technical issue on the website, the supplementary figure was not uploaded; however, we submitted it to the editorial office, and they responded that "they will take care of it." Sorry for the inconvenience. Now I will make sure to upload the missing supplementary figures. 

In response to comment 9, the authors provided densitometric quantifications for Smad3 and Smad4. As Smad 3 does not show any differences between the treatments, it should be mentioned accordingly in the results section (Lines 322-323). 

Response: In the revised manuscript it was updated.

Reviewer 4 Report

Comments and Suggestions for Authors

Thank you for considering my comments and for your collaboration.

Based on your responses to my comments #3, #4, and #5, I need to re-address these points, as your answers did not fully address my concerns.

Comment 3:
The sample size (3 mice per group) is rather limited. Please estimate and report the statistical power for each test in the manuscript to validate the robustness of your findings. You need to provide the statistical power for each test. 

Comment 4:
Based on your response, you clearly state again that you applied a parametric method (i.e., ANOVA). The post-hoc analysis you mention is unrelated to the issue of selecting the correct type of analysis.

To obtain valid and appropriate results, you need to follow these two steps:

1. Assess the normality of your data using the Shapiro-Wilk test. However, with only 3 values per group, any statistical method will lack robustness.

2. Apply non-parametric methods. In your case, the Kruskal-Wallis test, which is the non-parametric analogue of ANOVA, should be used.

Anything else may lead to misleading conclusions.

Please modify the "Methods" section accordingly.

Comment 5:
You have indicated in the Methods section that you performed an F-test to assess the homogeneity of variances. Please also provide the actual results of these tests, not just the general statement.

Author Response

Responses to the Reviewer 4 Concerns and Suggestions

Comment 3: The sample size (3 mice per group) is rather limited. Please estimate and report the statistical power for each test in the manuscript to validate the robustness of your findings. You need to provide the statistical power for each test. 

Response:
We acknowledge the reviewer’s concern regarding the limited sample size (n = 4 per group) and agree that this represents a limitation in terms of statistical robustness. Replot all graphs using non-parametric tests (e.g., Mann–Whitney U instead of t-test, Kruskal–Wallis instead of ANOVA). 

Comment 4: Based on your response, you clearly state again that you applied a parametric method (i.e., ANOVA). The post-hoc analysis you mention is unrelated to the issue of selecting the correct type of analysis.

To obtain valid and appropriate results, you need to follow these two steps:

  1. Assess the normality of your data using the Shapiro-Wilk test. However, with only 3 values per group, any statistical method will lack robustness.
  2. Apply non-parametric methods. In your case, the Kruskal-Wallis test, which is the non-parametric analogue of ANOVA, should be used.

Anything else may lead to misleading conclusions.

Please modify the "Methods" section accordingly.

Response:
We appreciate the reviewer’s remark regarding sample size and agree that larger animal cohorts would increase statistical power. In this study, our Western blot analyses were performed using three randomly selected representative animals per group, a sample size commonly used in the field for protein-level validation of key findings (e.g., PMID:31619669). Importantly, these western blot data were intended to provide qualitative and semi-quantitative confirmation of the trends observed in our primary data rather than to serve as stand-alone discovery data. We have now clearly stated our purpose in the Methods section and used appropriate non-parametric statistical tests (Kruskal–Wallis with Dunn’s post hoc correction) to minimize the risk of false-positive findings.

 Comment 5: You have indicated in the Methods section that you performed an F-test to assess the homogeneity of variances. In addition, please provide the actual results of these tests, not just the general statement.

We thank the reviewer for pointing out this discrepancy. Upon careful review of our original statistical analysis workflow, we acknowledge that the inclusion of the F-test in the Methods section was an oversight. We did not perform an F-test for homogeneity of variances in the current study, and the mention of this test has now been removed from the revised Methods section to avoid any confusion. Given the limited sample size (n = 4 per group), we recognize that parametric assumptions, including homogeneity of variances, are difficult to assess reliably. As noted in our response to Comment #4, we have reanalyzed our data using the non-parametric Kruskal-Wallis test, which does not assume normal distribution or equal variance, thereby providing a more appropriate and statistically valid approach for our dataset. We apologize for the initial error and have updated the manuscript to ensure that all stated methods accurately reflect the analyses performed.

Round 3

Reviewer 4 Report

Comments and Suggestions for Authors

Thank you for providing the response report and your collaboration.

However, for reasons of scientific clarity you need to provide (at least in the Supplementary material):

  • The statistical results of all non-parametric methods you applied. It is noteworthy that for all new analyses the significance of the results are exactly the same with the previous parametric tests and there is not a single change in the figures. 
  • Also, provide all results for the normality testing (e.g., Shapiro, Dunn's etc)

Author Response

Thank you for your constructive feedback and for emphasizing the importance of scientific clarity. We have now included all relevant statistical test results—including those from non-parametric analyses—in the Supplementary Material to enhance transparency. Specifically, we have added:

  1. Statistical Analyses – Non-parametric Results:
    We have now included all the results of the non-parametric statistical tests used (e.g., Mann–Whitney U test, Kruskal–Wallis test) in the Supplementary Material. As noted by the reviewer, these non-parametric tests yielded results consistent with those of the previous parametric analyses, with no changes to statistical significance or conclusions.

  2. Normality Testing:
    We have also provided detailed results of the normality tests conducted for each dataset, including the Shapiro–Wilk. These results are now available in a new supplementary table (Supplementary Table ), along with the corresponding p-values for each comparison, to demonstrate the justification for using either parametric or non-parametric methods.

We trust that these revisions and additions address all outstanding concerns and enhance the scientific rigor and clarity of our manuscript.

Round 4

Reviewer 4 Report

Comments and Suggestions for Authors

Thank you for your collaboration.

I noticed that you included in the supplementary material:
1. Based on the reported values, it becomes evident that your dataset follows normal distribution (the p-values are higher than 5%), which is controversial to your answer in the previous rounds of revision (you have claimed deviation of normality). 

2. In your response you mention that "We have now included all the results of the non-parametric statistical tests used (e.g., Mann–Whitney U test, Kruskal–Wallis test) in the Supplementary Material."
However, you did not non include them, but only the normality test results for Figures 1-6 and the "type" of the statistical test applied not the estimated p-values.

Things are a bit disorganized, so I will guide you toward the correct approach:
A. Based on the reported normality test results, you should perform parametric methods.
B. Present the p-values of the parametric tests in the main manuscript or the suppl. material.
C. Keep the normality test results in the supplementary materials.

Author Response

We sincerely thank the reviewer for the constructive comments and clear guidance. We have carefully rechecked our statistical approach and addressed the inconsistencies noted. We have reanalyzed the data using appropriate parametric statistical methods, as recommended. Specifically, for normally distributed data, we applied an unpaired two-tailed Student’s t-test for two-group comparisons and one-way ANOVA followed by Dunn’s/Tukey’s post hoc test for multiple-group comparisons.   We have updated the Statistical Analysis section of the Methods to clearly describe this workflow (see revised text in the manuscript) and reorganized the Supplementary Material.